# Chronic Ingestion of Bicarbonate-Rich Water Improves Anaerobic Performance in Hypohydrated Elite Judo Athletes: A Pilot Study

**DOI:** 10.3390/ijerph18094948

**Published:** 2021-05-06

**Authors:** Jakub Chycki, Maciej Kostrzewa, Adam Maszczyk, Adam Zajac

**Affiliations:** Institute of Sport Sciences, The Jerzy Kukuczka Academy of Physical Education in Katowice, 40-065 Katowice, Poland; j.chycki@awf.katowice.pl (J.C.); m.kostrzewa@awf.katowice.pl (M.K.); a.maszczyk@awf.katowice.pl (A.M.)

**Keywords:** bicarbonate water, anaerobic performance, hypohydration, rehydration, combat sports

## Abstract

In combat sports, anaerobic power and anaerobic capacity determine sport performance and the dominant metabolic pathways. The decline in performance during exercise that is attributed to the cumulative effects of fatigue, including excessive accumulation of metabolites, depletion of energy substrates, and water and electrolyte disturbances, seems to be of greatest significance. In our experiment, we evaluated the effectiveness of three weeks of bicarbonate-rich water ingestion on anaerobic performance in a state of hydration and dehydration in elite judo athletes. Eight male, elite judo athletes participated in two single-blind, repeated-measures trials. They were assigned to two hydration protocols, ingesting low mineralized table water and bicarbonate-rich water. Anaerobic performance was evaluated by two 30 s Wingate tests for lower and upper limbs, respectively, under conditions of hydration as well as exercise-induced dehydration. Resting, post-ingestion, and post-exercise concentrations of bicarbonate (HCO_3_), urine osmolality (U_OSM_), urine specific gravity (U_GRAV_), and lactate (La) were measured. The current investigation assessed two related factors that impair anaerobic performance—hypohydration and buffering capacity. High-bicarbonate water ingestion improved buffering capacity, and we demonstrated the potential role of this mechanism and its phenomenon in masking the adverse effects of dehydration in the context of repeated high-intensity anaerobic exercise (HIAE).

## 1. Introduction

In combat sports, anaerobic power and anaerobic capacity determine sport performance and the dominant metabolic pathways. During competition, which may last from 4 min of regular time to even 7 min real time of fighting in judo, a decrease in repeated explosive power is primarily attributed to the decompensated acid–base balance, hypohydration, and depletion of muscle glycogen [1]. Due to the indefinite duration of sports competition, it is difficult to indicate the dominant mechanism responsible for impairment of anaerobic metabolism. Factors determining fatigue are complex and include both central and peripheral components [1,2]. The decline in performance during exercise is attributed to the depletion of substrates and the excessive accumulation of metabolites, such as hydrogen ions (H), potassium (K), and phosphate ions (Pi). Additionally, water and electrolyte disturbances seem to affect exercise efficiency and fatigue [3,4,5]. These phenomena are compounded by the growing uncompensated water loss, leading to hypohydration [6]. It is unclear how much hypohydration impairs anaerobic performance, and inconsistent results may be due to the use of different protocols to induce hypohydration and different methods of evaluating strength and power abilities. Some authors have observed significant reductions in isometric and isokinetic force after 2% hypohydration [6], while others did not register significant changes in muscular power even after a 4% hypohydration [6,7,8]. Considering the above data, there are two potential strategies that may support anaerobic performance: control of hydration status and improvement of buffering capacity.

Various methods of assessing hydration status are used during monitoring of the training process. These include electrical impedance analysis, hematologic markers, isotope indices, and urinary markers. In our study, we used urine osmolality (U_osm_) and urine-specific gravity (U_grav_). This choice is consistent with the suggestion of Popowski et al. [9], who compared the validity of U_osm_ and U_sg_ with plasma osmolality and concluded that both are strongly correlated and are good measurements of the state of hydration.

Effective hydration determines proper functioning of the human body, at rest and especially during and after exercise. Water allows for homeostasis, facilitates most biochemical reactions, and allows numerous particles and compounds to dilute. It helps in the transport of metabolites and utilization of by-products [10,11]. Several comprehensive reviews of the effect of dehydration on local muscular endurance, strength, anaerobic capacity, jumping performance, and specific sport skills in team sports games have revealed negative effects of dehydration <2% body mass [6,12,13]. Exercise induces ionic changes within contracting muscles that contribute to the development of metabolic acidosis. The increased intramuscular acidity may then limit the ability to perform high-intensity exercise [2]. Hypohydration caused by sweat loss is associated with a reduction in blood plasma and total blood volume, which impairs cardiovascular function, muscle blood flow, and thermoregulatory function [2,14]. We thus hypothesize that the excess of metabolic end-products (namely, H^+^ and Pi), which disturb cellular homeostasis and muscle contraction, are more effectively transported under optimal hydration of the body and blood viscosity [13,15]. 

Given the relationship between exercise-induced acidosis and fatigue, the ingestion of potential buffering agents such as sodium bicarbonate has been suggested to attenuate metabolic acidosis and improve anaerobic performance, as has been reported in our previous research [8,14,16,17]. The ergogenic effects of bicarbonate have been confirmed previously [8,14]. Bicarbonate increases extracellular pH and enhances the efflux of lactate and H^+^ from muscle cells [18]. Buffering of protons can attenuate changes in pH and enhance the muscles’ buffering capacity. The indicated mechanism allows for a greater amount of lactate to accumulate in the muscle cell, thus increasing glycolytic capacity [19]. There is also evidence that the ergogenic effect of bicarbonate is more pronounced during repeated HIAE than during sustained continuous exercise [16]. 

As a consequence of these scientific data and our previous research, we have assumed that bicarbonate-rich water ingestion can improve hydration and anaerobic performance, i.e., (a) reduce dehydration during exercise, (b) accelerate re-hydration, and (c) increase the efficiency of high-intensity anaerobic exercise (HIAE). Given the limited research on well-trained athletes and the possible effect of bicarbonate intake on the relationship between hypohydration and anaerobic performance, this study aimed to determine the effects of buffering capacity on lactate efflux and selected Wingate test variables—total work and fatigue slope under different hydration conditions. We hypothesized that the ergogenic effect of HCO_3_^−^, which is based on increased buffering capacity, will allow for a greater volume of high-intensity exercise, both while hydrated, after exercise-induced hypohydration, and following rehydration, which may have an indirect influence on masking adverse effects of dehydration in repeated HIAE.

## 2. Materials and Methods 

### 2.1. Participants 

Eight male elite judo athletes were enrolled in the study. All participants had at least twelve years of training experience, international sports level, and they were members of the Polish National Team. They were all medalists of the National Championships and participants of the World Cup. They constituted a homogenous group in regard to age, training experience, somatic characteristics, and aerobic and anaerobic fitness. To determine the sample size, we used LEO Sound^®^ software (LEO Sound, Toronto, Canada) at a 95% confidence level with a 6.5% margin of error and standard deviation, given that population size is 28. The size of the population includes members of the Polish National Judo Team in weight categories of 73 kg and 81 kg in 2019–2020. Only these weight categories were included, to create a homogenous group of participants. Including more athletes from extreme weight categories (under 60 or above 100 kg) would make the hydration dehydration protocols very difficult. The sample size was estimated at *n* = 8. A brief description of the participants’ characteristics is given in Table 1. A total of eight men participated in two single-blind, repeated-measures trial. They were assigned to two hydration protocols, ingesting low mineralized table water and bicarbonate-rich water (Figure 1.). All participants had valid medical examinations and showed no contraindications to participate in the study. The participants were informed verbally and in writing about the experimental protocol and the possibility to withdraw from the research at any stage of the experiment. The study was approved by the Research Ethics Committee at the Academy of Physical Education in Katowice, Poland (ethic reference KB-4/2020), and conformed to the principles of the Declaration of Helsinki.

### 2.2. Experimental Design

The experiment consisted of two interventions—hydration protocols (trials)—during which three series of laboratory analyses were performed. The tests were carried out at the stage of preparation for intervention—hydration assessment and verification (S0)—after the first hydration protocol, 21 days after table water ingestion (S1), and after 21 days of ingestion of bicarbonate-rich water (S2). The research was carried out in the within-subjects design model. All participants participated in the first and second trials (Figure 1). The study was conducted during the preparatory period of the annual training cycle, when a high volume of work dominated the daily training loads. The participants refrained from exercise for two days before testing to minimize the effects of fatigue.

#### 2.2.1. Screening Tests

All selected participants for the current study underwent preliminary tests and medical examinations before the intervention. The main criteria for selection included sports performance at the National Championships and the World Cup in the last 24 months. Moreover, for the homogeneity of the research group, the authors used additional criteria: VO_2max_, higher than 58.0 mL/kg/min; TW, upper limbs, higher than 180 (J/kg); TW, lower limbs, higher than 220 (J/kg). The participants were excluded if they reported a history of cognitive deficiencies or were taking neuroactive or psychoactive drugs, stimulants, antioxidants, or other illegal substances. The inclusion and exclusion criteria were assessed at the screening stage, based on an interview, health assessment, and analysis of fitness test results. 

Six of the eight participants in the experiment had performance tests (VO_2max_, Wingate Test) that were performed 3 weeks prior to the project. In the case of two participants, the performance tests were carried out at the qualification stage, one week before the intervention. These discrepancies were caused by participation in competitions. Afterwards, each study participant visited the laboratory to get acquainted with the research protocol. 

#### 2.2.2. Diet and Hydration Protocol

Energy consumption, as well as macro- and micronutrient intake of all subjects, was determined by the 24 h nutrition recall 3 weeks prior to study initiation. The participants were placed on an isocaloric (3455 ± 436 kcal/d) mixed diet (55% carbohydrates, 20% protein, 25% fat) prior to and during the investigation. The pre-trial meals were standardized for energy intake (600 kcal) and consisted of carbohydrates (70%), fat (20%), and protein (10%). The participants did not take any medications, supplements, or ergogenic substances for 3 weeks before and during the study. During the experiment, the study participants lived in the dormitory and were fed at the academy cafeteria. The meals were prepared in the form of 24 h menus for seven days of the week. All meals were planned and supervised by a nutritionist. The quality and quantity of the food products were strictly controlled, maintaining proper proportions between the major macronutrients. 

The volume of water intake was individualized based on the recommendation of the National Athletic Trainers Association and averaged 3.2–3.4 L per day. The bicarbonate-rich water contained 5700 mg/dm^3^ of permanent ingredients, which is classified as high mineral content. The bicarbonate ion HCO_3_^−^ (1300–4250 mg/dm^3^) consisted of the dominant anion, while sodium (Na^+^ 1200–1700 mg/dm^3^) dominated among cations. The water contained 50–75 mg/dm^3^ Ca^2+^, 35–55 mg/dm^3^ Mg^2+^, and 260–380 mg/dm^3^ Cl^−^). A comparison of the composition of the waters used in the experiment is presented in Table 2.

#### 2.2.3. Experimental Protocol—Hydration, Dehydration, Rehydration

During the two days of laboratory analysis, the same tests were performed. The participants reported to the laboratory at 08:00 or 8:30 with morning urine samples. Urine-specific gravity of 1020 (kg/L) confirmed that the participants began the tests in a euhydrated state. Then, they underwent medical examinations and somatic measurements. The measurements of body mass were performed on a medical scale with a precision of 0.1 kg (Seca, mo.635, Seca Co., Hamburg, Germany). Body composition was evaluated using the electrical impedance technique (Inbody 720, Biospace Co., Tokyo, Japan). After sitting for 10 min in a 22 °C environment, a blood sample was collected. Lactate and acid–base balance variables were determined from 1 mL capillary blood samples. 

At 9:00, they were subjected to the dehydration protocol, which consisted of continuous running on a treadmill at an intensity of 50% VO_2max_. During exercise-induced dehydration, the participants were weighed at 12 min intervals until they reached a 3% weight loss. At that moment, they were considered dehydrated. Body mass was measured (±50g) during exercise, while participants briefly stepped off the treadmill onto a floor scale. The measurement was carried out in underwear. The control measurements did not last longer than 45 s. The duration of exercise was supervised and recorded. Drinking any liquid during and after exercise was prohibited. Under dehydration, anaerobic performance, hydration status, and biochemical variables were evaluated.

The rehydration protocol was carried out between 12 and 15 o’clock. The participants used table or bicarbonate-rich water by drinking the recommended volume (VR) every 15 min. The VR was 1/12 volume of water lost. The protocol lasted until one of two criteria were reached: (1) fluids were replenished, or (2) 150% of the lost volume of water was drunk. The volume and frequency of urine output, U_osmol,_ U_SG,_ and U_pH_ were evaluated (Figure 2).

### 2.3. Biochemical Assays

#### 2.3.1. Lactate, Acid-Base Balance, Ion Concentration

To determine lactate concentration (LA), acid-base equilibrium and electrolyte status the following variables were evaluated: LA (mmol/L), blood pH, pCO_2_ (mmHg), pO_2_ (mmHg), HCO_3_^−^ _act_ (mmol/L), HCO_3_^−^_std_, (mmol/L), BE (mmol/L), O_2SAT_ (mmol/L), ctCO_2_ (mmol/L), Na^+^ (mmol/L), K^+^ (mmol/L), and Ca^2+^ (mmol/L). The measurements were performed from fingertip capillary blood samples at rest and after 3 min of post-exercise recovery. Determination of LA was based on an enzymatic method (Biosen C-line Clinic, EKF-diagnostic GmbH, Barleben, Germany). The remaining variables were measured using a Blood Gas Analyzer GEM 3500 (GEM Premier 3500, Breda, Netherlands).

#### 2.3.2. Hydration Status

Urine samples were taken at the stage hydration assessment and verification (S0) and at the end of the hydration protocols (S_1,_ S_2_), at rest (hydrated), as well as under dehydrated conditions and after rehydration. They were placed in a plastic container and mixed with 5 mL/L of 5% solution of isopropyl alcohol and thymol for preservation. Urine samples were assayed for the presence of blood and proteins. Specific gravity was determined using the Atago Digital refractometer (Atago Digital, Tokyo, Japan). Urine pH was determined based on the standardized Mettler Toledo potentiometer (Mettler Toledo, Zaventem, Belgium).

#### 2.3.3. Anaerobic and Aerobic Performance 

Anaerobic performance was evaluated by two 30 s Wingate tests for lower and upper limbs, with a passive rest interval of 3 min between the bouts of exercise. The test was preceded by a 20 min general warm-up, as well as a 5 min specific warm-up. The preparations consisted of a 12 min, low-intensity, continuous exercise on an elliptical cross trainer (Keiser Stride, m5i, Fresno, CA, USA) and 8 min of general calisthenic exercises. Then, the participants performed specific exercises on diagnostic ergometers—5 min with a resistance of 100 W and a cadence of 70–80 rpm for the lower limbs and 40 W and 50–60 rpm for the upper limbs. Following the warm-up, the test began, in which the objective was to reach the highest cadence in the shortest possible time and to maintain it throughout the trial. The lower limb Wingate protocol was performed on an Excalibur Sport ergocycle with a resistance of 0.8 Nm·kg^−1^ (Lode BV, Groningen, The Netherland). The upper body Wingate test was carried out on a rotator with a load of 0.45 Nm·kg^−1^ (Brachumera Sport, Lode, The Netherland). Each subject completed the test trials with incomplete rest intervals. The variables of peak power P_max_ (W/kg), mean power P_mean_ (W/kg), and total work performed W_t_ (J/kg) were registered and calculated by the Lode Ergometry Manager. To evaluate the level of fatigue during the anaerobic test, the fatigue slope (F_slope_) was calculated as the rate of decline in power during the 30 s test, expressed in W·s^−1^ (LEM, software package, Groningen, The Netherlands). 

The ramp VO_2max_ test was performed on a treadmill (H/P Cosmos, Pulsar, Germany), starting at a speed of 6 km/h, which was increased linearly (1_km/h_/1_min_) until volitional exhaustion. During the test, heart rate (HR), oxygen uptake (VO_2_), expired carbon dioxide (CO_2_), minute ventilation (VE), breath frequency (BF), and respiratory exchange ratio (RER) were measured continuously using the MetaLyzer 3B-2R spiroergometer (Cortex, Leipzig, Germany) in the breath-by-breath mode. The following two criteria were used to determine VO_2max_: (a) a plateau in VO_2_ despite an increase in running speed, (b) RER > 1.10.

### 2.4. Statistical Analysis

The Shapiro–Wilk, Levene, and Mauchly’s tests were used to verify the normality, homogeneity, and sphericity of the sample’s data variances, respectively. Verifications of the differences between the considered values before and after three weeks of bicarbonate-rich water ingestion, between hydration, hypohydration, and rehydration conditions were verified using ANOVA with repeated measures. Statistical significance was set at *p* < 0.05. All statistical analyses were performed using Statistica 9.1 (TIBCO Software Inc., Palo Alto, California, CA, USA) and Microsoft Office (Microsoft, Redmont, Washington, DC, USA), and are presented as means with standard deviations.

## 3. Results

All participants completed the described test protocols. The results of ANOVA revealed significant differences between 21 days of table water and bicarbonate-rich water ingestion. The following variables were considered:

### 3.1. Anaerobic Performance, Lactate Concentration, and Acid–Base Balance

Tests revealed a statistically significant increase in Upper Limb Total Work in hypohydration conditions (from 161.67 J/kg to 195.43 J/kg with *p* = 0.014) after bicarbonate-rich water ingestion. The analysis also showed statistically significant increases in values for Upper Limbs Mean Power (from 6.56 W/kg to 7.79 W/kg with *p* = 0.007) in the hypohydration state after bicarbonate-rich water ingestion. In contrast, after table water ingestion (following the hydration standardization protocol), no statistically significant results in either hydration and hypohydration status were shown (Table 3).

We also revealed a statistically significant increase in post-exercise LA concentration (from 15.20 mmol/L to 16.95 mmol/L with *p* = 0.049—hydration conditions, and from 13.46 mmol/L to 15.94 mmol/L with *p* = 0.012—hypohydration) after bicarbonate-rich water ingestion. 

Additionally, a significant increase in blood HCO_3_^−^ at rest (in both hydration and hypohydration status: Hydration—from 27.37 mmol/L to 28.91 mmol/L with *p* = 0.012, Hypohydration—from 26.34 mmol/L to 27.96 mmol/L with *p* = 0.001) was observed in the athletes after 21 days of bicarbonate-rich water ingestion (Table 4). 

### 3.2. Hydration

The results of ANOVA with repeated measures showed statistically significant differences in urine osmolality (U_OSM_) between hydration (*p* = 0.036), hypohydration (0.042) condition, and rehydration, both during the first (60 min: *p* = 0.05) and second phase of rehydration (120 min: *p* = 0.003, and 180 min: *p* = 0.026) when after 21 days table water and 21 days of bicarbonate-rich water ingestion values were compared (Table 5). 

Other significant changes occurred in the volume of urine output (excretion) (from 484 mL to 316 mL with *p* = 0.037) (Table 6).

## 4. Discussion

The main objective of the present study was to evaluate the effects of three weeks of bicarbonate-rich water ingestion on selected anaerobic performance variables in a state of full hydration and after exercise-induced dehydration in elite judo athletes. This is the first study to investigate the effectiveness of water with high NaHCO_3_^−^ content on anaerobic performance and rehydration in elite athletes undertaking acute, exercise-induced dehydration. First, we would like to justify the choice of the study design. The lack of a control group is caused by the very demanding inclusion criteria. In this context, we want to emphasize that the athletes were members of the Polish Judo National Team and competed at the highest level of national and international competitions. Clarification is also needed of the Wingate test not being performed after rehydration. This was due to a very demanding and long-lasting research protocol. The authors decided that performing the HIAE protocol six times would be extremely difficult for the athletes. 

### 4.1. Anaerobic Performance and Lactate Metabolism

The current investigation demonstrated a significant improvement in anaerobic performance of athletes after 3 weeks of bicarbonate-rich water ingestion. The findings of this study are in agreement with McNaughton [16,20], who reported significant improvements in ergometer performance following NaHCO_3_^−^ supplementation and many other controlled studies in various sports disciplines [2,8,14,21]. The ergogenic effect of sodium bicarbonate on exercise performance stems from the reinforced extracellular bicarbonate buffering capacity to regulate acid–base balance during high intensity exercise [21,22]. Elevated levels of bicarbonate enlarge the gradient between extracellular and intracellular H^+^, which stimulates the lactate/H^+^ cotransporter [23]. 

In agreement with our previous work [8,14], greater blood lactate was observed post exercise after three weeks of bicarbonate­-rich water ingestion when compared with table water, suggesting that NaHCO_3_^−^ assists lactate efflux from muscle. An increased blood HCO_3_^−^ and the accompanying increase in blood buffering capacity would, in turn, help maintain the pH gradient between the cell and blood. This would lead to enhanced H^+^ efflux from the cell (confirmed by our data) with the rise in blood La concentration being a consequence of greater H^+^/La cotransport.

### 4.2. Hydration Status 

While exogenous HCO_3_^−^ is recognized to increase the efficiency of anaerobic performance by influencing buffering capacity, it has additionally been shown that the sodium supports loading fluid, confirming its ergogenic potential by expanding plasma volume [24,25]. Increased plasma volume is part of a pre-competition strategy used by athletes who are rapidly dehydrating, including fluid restriction or exposition to uncompensated water loss during competition [26]. The use of water with a high Na^+^ and HCO_3_^−^ content may offer a simple solution of increasing Na^+^ intake to optimize effective rehydration while enhancing blood buffering capacity. Considering the above, it seems justified to co-ingest a required volume of fluid, with an effective dose of NaHCO_3_^−^ (0.3–0.4 g of NaHCO_3_^−^ per kilogram BM) [27]. The well-tolerated volume of 1500 mL of fluid in dehydration along with the ergogenic strategy of using NaHCO_3_^−^ provides the recommended dose of Na^+^ for rehydration strategy (170 mmol of Na^+^∙L^–1^) [22,23,27]. It has been suggested that the contribution of sodium and fluid associated with NaHCO_3_^−^ ingestion may be implicated in its reported benefits on exercise performance and hydration status control. Mitchell et al.’s [28] results suggest that both NaHCO_3_^−^ and placebo supplementation improved high-intensity exercise performance when sodium content (154 mmol∙L^–1^) and fluid volume (~1500 mL) were controlled. The results of these studies are in line with the presented acceleration of rehydration following exercise-induced dehydration. 

We are aware of numerous limitations of this study. First of all, we examined the hydration state using commonly available methods, based on monitoring urine osmolality and urine-specific gravity, and were aware of their limitations. This fact did not allow us to unequivocally assess the status of hydration at sensitive evaluation points, i.e., qualification for research, after exercise-induced dehydration, and after rehydration. Measuring hydration status is challenging due to complex dynamics associated with fluid regulation. Water balance is a continuous process of water loss from different compartments of the body. Assessment techniques aim to measure one or more fluid compartments either directly or indirectly. For this reason, we intend to use the isotope method of hydration assessment in future projects. Despite the small, yet highly representative, research sample of elite judo athletes of two weight categories, we observed the same single-line biological changes. We will confirm this on a larger sample of athletes. Future research will also include a study of exercise metabolites in different states of hydration to provide a deeper understanding of the biological mechanisms related to this intervention.

## 5. Conclusions

The results of our experiment are in line with many other well-controlled studies, which used high-intensity exercise protocols to verify the effectiveness of ergogenic substances that improve buffering capacity, including HCO_3_^−^. However, there are several novelties to our study, which should be addressed. First, we conducted research on elite combat sports athletes. In addition, this is the first study to assess two related factors that impair anaerobic performance—hypohydration and buffering capacity. Certainly, anaerobic performance is affected by the hydration status of the athletes. Both in the context of competition and training routines, hydration is part of the strategy to reduce the possibility of uncompensated water loss. The presented results show the benefits of high-bicarbonate water ingestion on buffering capacity, anaerobic fitness, and the rate of rehydration. The athletes in the present study used bicarbonate-rich water to improve their buffering capacity, and we demonstrated the potential role of this mechanism and its phenomenon in masking the adverse effects of dehydration in the context of repeated HIAE. 

## Figures and Tables

**Figure 1 ijerph-18-04948-f001:**
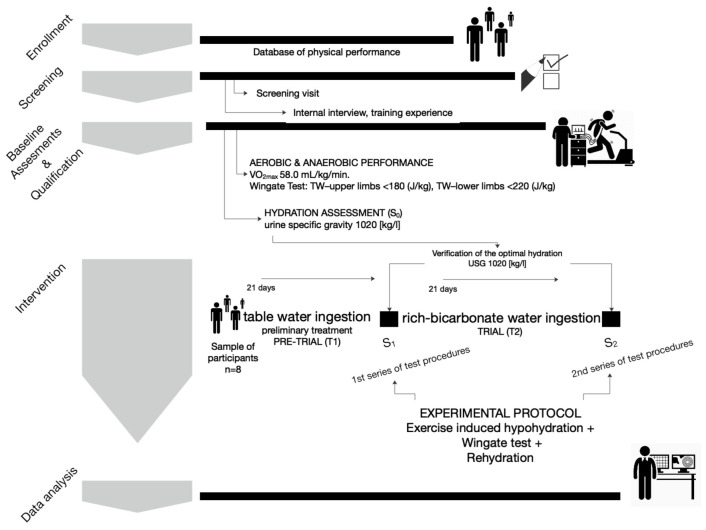
Flowchart.

**Figure 2 ijerph-18-04948-f002:**
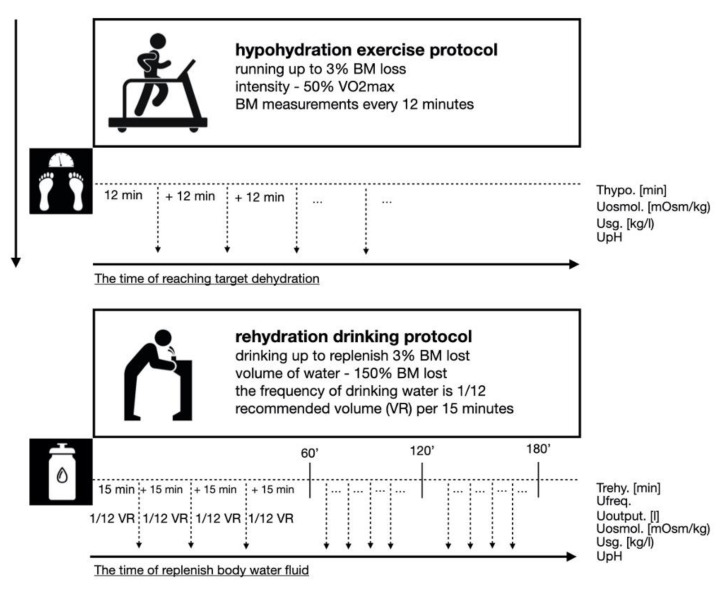
Schematic presentation of the dehydration and rehydration protocols.

**Table 1 ijerph-18-04948-t001:** Characteristics of the study participants.

Variables	Participants (*n* = 8) Mean ± SD
Age (yrs.)	24.3 ± 0.5
Height (cm)	181 ± 2.3
Body mass (kg)	81 ± 2.4
TW_–upper limbs_ (J/kg)	192.10 ± 5.50
TW_–lower limbs_ (J/kg)	246.00 ± 6.50
MP_–upper limbs_ (W/kg)	7.22 ± 0.47
MP_– lower limbs_ (W/kg)	9.14 ± 0.87
VO_2max_ (ml/kg/min)	59.7 ± 3.2

Note: TW—Total Work; MP—Mean Power.

**Table 2 ijerph-18-04948-t002:** Chemical properties of water used in the study.

Variable	Measurement Unit	Bicarbonate-Rich Water Mean ± SD	Table Water Mean ± SD
HCO_3_^−^	mg/dm^3^	4002.02 ± 120.3	3.62 ± 0.12
Cl^−^	mg/dm^3^	264.00 ± 0.12	0.41 ± 0.03
SO_4_^2−^	mg/dm^3^	<50	1.60 ± 0.09
Na^+^	mg/dm^3^	1154.30 ± 134.0	1.21 ± 0.05
K^+^	mg/dm^3^	32.47 ± 4.56	0.30 ± 0.03
Ca^2+^	mg/dm^3^	62.00 ± 4.00	1.21 ± 0.05
Mg^2+^	mg/dm^3^	52.20 ± 0.04	0.40 ± 0.04

Note: Data shows mean values ± SD of three analyses of each type of water.

**Table 3 ijerph-18-04948-t003:** The differences in anaerobic performance variables—upper limbs, after table water, and bicarbonate-rich water ingestion.

Variables		Hydration	Hypo-Hydration
		Mean ± SD	Mean ± SD
TW/kg (J/kg)	Table water	188.04 ± 16.63	161.67 ± 19.18
	Bicarbonate water	201.22 ± 21.65	195.43 ± 24.61 #
MP/kg (W/kg)	Table water	7.63 ± 0.58	6.56 ± 0.70
	Bicarbonate water	8.13 ± 1.11	7.79 ± 0.71 #
F_s_ (W·s^−1^)	Table water	27.38 ± 6.63	26.96 ± 5.40
	Bicarbonate water	27.53 ± 6.31	22.97 ± 3.9
PP/kg (W/kg)	Table water	12.11 ± 2.65	11.02 ± 3.24
	Bicarbonate water	12.98 ± 2.92	11.9 ± 2.72

Note: MP—Mean Power; F_s_– Fatigue Slope—rate of power decrease in watts per second, T—Total Work, PP—Peak Power; #—statistically significant differences with *p* < 0.05 between table water vs. bicarbonate-rich water.

**Table 4 ijerph-18-04948-t004:** The differences in post-exercise blood plasma lactate concentration, as well as the resting concentration of bicarbonate and blood pH values after table water and bicarbonate-rich water ingestion.

Variables		Hydration	Hypo-Hydration
		Mean ± SD	Mean ± SD
LA_max_ (mmol/L)	Table water	15.20 ± 1.63	13.46 ± 1.73
	Bicarbonate water	16.95 ± 1.97 #	15.94 ± 2.30 #
HCO_3_^−^ _rest_ (mmol/L)	Table water	27.37 ± 0.07	26.34 ± 0.09
	Bicarbonate water	28.91 ± 0.09 #	27.96 ± 0.08 #
pH (−log[H+])	Table water	7.43 ± 0.003	7.42 ± 0.003
	Bicarbonate water	7.44 ± 0.005	7.43 ± 0.005

Note: LA_max_—post-exercise blood plasma lactate concentration; HCO_3_^−^ bicarbonate; #—statistically significant difference with *p* < 0.05 between table water vs. bicarbonate-rich water.

**Table 5 ijerph-18-04948-t005:** The differences in hydration variables after table water and bicarbonate-rich water ingestion.

Variables		Hydration	Hypo-Hydration	Re-Hydration 60 min	Re-Hydration 120 min	Re-Hydration 180 min
		Mean ± SD	Mean ± SD	Mean ± SD	Mean ± SD	Mean ± SD
U_OSM_ (mOsm·L^−1^)	Table water	769.62 ± 55.82	913.125 ± 53.48	644.62 ± 42.92	512.87 ± 67.34	192.12 ± 50.50
	Bicarbonate water	679.87 ± 57.92 #	862.62 ± 51.05 #	568.37 ± 38.25 #	378.87 ± 69.01	137.37 ± 52.63
U_pH_ (−log[H^+^])	Table water	5.43 ± 0.82	5.37 ± 0.44	5.25 ± 0.45	6.12 ± 1.27	7.68 ± 0.99
	Bicarbonate water	7.12 ± 0.44	7.12 ± 0.64	7.37 ± 0.79	8.56 ± 0.417	8.5 ± 0.5
U_SG_ (kg/L)	Table water	1.011 ± 0.002	1.024 ± 0.005	1.017 ± 0.007	1.015 ± 0.008	1.004 ± 0.003
	Bicarbonate water	1.005 ± 0.002	1.018 ± 0.004	1.015 ± 0.004	1.007 ± 0.005	1.002 ± 0.0009
BM (kg)	Table water	81.0 ± 2.4	78.5 ± 1.8	80.1 ± 1.6	80.9 ± 0.6	-
	Bicarbonate water	81.5 ± 1.6	79.0 ± 1.2	81.4 ± 0.8	81.7 ± 0.8	-

Note: U_OSM_—urine osmolality; U_pH_—urine pH; U_SG_—urine-specific gravity; BM—body mass; #—statistically significant differences with *p* < 0.05 between table water vs. bicarbonate-rich water.

**Table 6 ijerph-18-04948-t006:** The differences in selected variables of hydration kinetics after table water and bicarbonate-rich water ingestion.

Variables		Mean ± SD
U_OUTPUT-V_ (mL)	Table water	484.00 ± 60.00
	Bicarbonate water	316.00 ± 45.00 #
Time_HYPO_ (mins)	Table water	42.00 ± 15.00
	Bicarbonate water	60.00 ± 15.00

Note: U_OUTPUT-V_—urine output volume during rehydration; U_OUTPUT-F_—urine output frequency during rehydration; Time_HYPO_—time to hypohydration (3%BM), #—statistically significant differences with *p* < 0.05 between table water vs. bicarbonate-rich water.

## Data Availability

The data presented in this study are available on request from the corresponding author.

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
