# Peer review of "Chronic Ingestion of Bicarbonate-Rich Water Improves Anaerobic Performance in Hypohydrated Elite Judo Athletes: A Pilot Study"

_ijerph, 2021, doi:10.3390/ijerph18094948_

Round 1

Reviewer 1 Report

The research presented in the manuscript has some shortcomings, but I believe given the elite nature of the participants, will still be of interest to readers. 

Overall, I think the manuscript type should be titled a 'case report', not a research 'article'.  

Specific feedback includes:

Introduction

Define HIAE...not all readers of this journal know this acronym. In the introduction, it may be useful to add something about the weigh in routine common in many combat sport and why you chose this dehydration/rehydration strategy. How does your strategy align with elite competition practices? This would be of interest to many.

Materials and methods

A sample size calculation has been added and estimated 28 participants required. The study includes 8 participants. There is no commentary why such a reduced number. I appreciate the elite nature of the participant group, and for this reason I strongly suggest the authors change this to a case report and remove the sample size calculation, or make mention of the elite nature of the participants, thus preventing a larger participant group.

Table 1, 2 and 3 require significant reformatting. Change paragraph centering to left margin, not center. Remove the double spacing. Fix the table formatting to remove word from either side of the tables. 

Were all the participants males? This is an important detail that I cannot find.

Can the authors better articulate the research design. I still don't quite understand the table water to bicarbonate rich water comparison in the results. 

Author Response

Response to reviewer 1

Comments and Suggestions for Authors

The research presented in the manuscript has some shortcomings, but I believe given the elite nature of the participants, will still be of interest to readers. 

We appreciate this comment. In the revised manuscript, we have modified the original version based on the comments and suggestions.

Comment 1:

Introduction.

Define HIAE...not all readers of this journal know this acronym. In the introduction, it may be useful to add something about the weigh in routine common in many combat sport and why you chose this dehydration/rehydration strategy. How does your strategy align with elite competition practices? This would be of interest to many.

Response 1:

Thank you for your comment. The abbreviation has been defined. Moreover, the authors justified the choice of the exercise protocol.

Comment 2:

Materials and methods.

I think the manuscript type should be titled a 'case report', not a research 'article'.

A sample size calculation has been added and estimated 28 participants required. The study includes 8 participants. There is no commentary, why such a reduced number. I appreciate the elite nature of the participant group, and for this reason I strongly suggest the authors change this to a case report and remove the sample size calculation or make mention of the elite nature of the participants, thus preventing a larger participant group.

Response 2:

We thank the reviewer for the suggestion. The indicated issue requires clarification. The  manuscript was inaccurate and incomplete. This excerpt showed the total population of elite judokas available (28), not a representative sample of the population (n=8). The authors introduced changes and additions. “…To determine the sample size we used LEO Sound® at a 95% confidence level with a 6,5% margin of error and standard deviation, given that population size is 28. The sample size was estimated at n=8…” (Page 2, Lines 90-92).Additionally, in line with the reviewer’s suggestions, the authors emphasized the elitism of the participants and clarified the inclusion criteria. Estimates were made by the authors as below:

To determine the sample size we used LEOSound® at a 95% confidence level with a 6,5% margin of error and standard deviation, given that population size is 28, we used the following formula:

where,

  • N = Population size,
  • Z = Critical value of the normal distribution at the required confidence level,
  • p = Sample proportion,
  • e = Margin of error

In this experiment

  • N = 28
  • Z = 1.95
  • p = 0.05 (uncertain sample proportion),
  • e = 6,5% or 0.065

n= (28 * (1.95 2)*0.05*(1-0.05)/(0.065 2)/(28 – 1+((1.95 2)* 0.05*(1-0.05)/(0.065 2))))=8

Sample Size = 8

Explanation of the Sample Size Formula

The calculation of the sample size formula has been done by using the following steps:

Step 1: Firstly, we determined the population size which is the total number of distinct entities in population and it is denoted by N. [Note: N=28]

Step 2: Next, we determined the critical value of the normal distribution at the required confidence level (critical value at 95% confidence level - 1.95)

Step 3: Next, we determined the sample proportion which can be used from previous survey results or be collected by running a small pilot survey. [Note: p=0.05.]

Step 4: Next, we determined the margin of error which is the range in which the true population is expected to lie. [Note: e=0.065]

Step 5: Finally, the sample size equation was derived by using population size (step 1), the critical value of the normal distribution at the required confidence level (step 2), sample proportion (step 3) and margin of error (step 4) as shown below.

The sample size was estimated at n=8.

“…To determine the sample size we used LEO Sound® at a 95% confidence level with a 6,5% margin of error and standard deviation, given that population size is 28. The sample size was estimated at n=8…” (Page 2, Lines 90).

Comment 3:

Table 1, 2 and 3 require significant reformatting. Change paragraph centering to left margin, not center. Remove the double spacing. Fix the table formatting to remove word from either side of the tables. 

Response 3:

The authors apologize for the inconvenience in the first version of the manuscript. Tables have been formatted.

Table 1. (Page 3, Lines 104), Table 2. (Page 5, Line 138), Table 3 (Page 7, Lines 222)

Comment 4:

Were all the participants males? This is an important detail that I cannot find.

Response 4:

This information should absolutely be exposed. It is true that the authors mention a group of men in paragraph 93,

“…A total of eight men participated in two, single-blind, repeated-measures trials. They were assigned to two hydration protocols, ingesting low mineralized table water and rich-bicarbonate water…” (Page 2, Lines 93). but the information was also added in the abstract and in the first sentences of the characteristics of the participants.“…Eight apparently male elite judo athletes were enrolled in the study…” (Page 2, Lines 86).

Comment 5:

Can the authors better articulate the research design. I still don't quite understand the table water to bicarbonate rich water comparison in the results. 

Response 5:

       At the beginning, the authors would like to mention that at the stage of constructing the experiment, several variants were analyzed and discussed. However, the single-blind, repeated measures trial model was chosen and considered to be justified. “…Eight male, elite judo athletes participated in two, single-blind, repeated-measures trials. They were assigned to two hydration protocols (Trials), ingesting low mineralized table water – pre-trial, and rich-bicarbonate water – trial…”                When attempting to justify the experimental model used, it should be emphasized that the main goal of the study is to compare selected anaerobic performance variables after exercise-induced dehydration. The participants of the experiment were elite athletes meeting the inclusion criteria, whose anaerobic performance is assessed in the state of hydration and dehydration (3% BM). Additionally, the rehydration time was observed. In this approach, the status of hydration, dehydration, and rehydration is a specific physiological state defined by weight loss and urine osmolality. Metabolic stress conditions, were achieved through: 1) hydration protocol 2) exercise leading to dehydration.

An important element of the research was to rule out the possibility that athletes are not optimally hydrated and that, as a consequence, the results of anaerobic performance assessment (double Wingate test protocol) in the state of hydration and subsequently in the state of exercise-induced dehydration will be false and unreliable.

Subsequently, the same participants - elite judo athletes - were subjected to two trials: 

  • PRE TRIAL- which is actually Preliminary Treatment. The purpose of this stage is the standardization of hydration conditions. Preliminary treatment included 21 days table water ingestion, re-evaluation of hydration status and qualification to the experimental protocol (exercise-induced dehydration + anaerobic performance tests + rehydration protocol).

 2)    TRIAL - the same scenario was used with rich-bicarbonate water ingestion. Two measurements: baseline and after 21 days of use rich-bicarbonate water were performed and analyzed.

Why did the authors use within-subjects designs model? First, because it is difficult to define table water as a therapeutic agent. Second, because the participants constituted an elite group of athletes with a limited number of participants. In the opinion of the authors, comparing the results of laboratory tests in the described procedure and stages of the experiment is fully justified and the methods of statistical analysis used should not be questioned.

The authors changed figure 1 to improve the readability of the study design …” (Page 4, Lines 114). We used the phrase “divided into” in an unfortunate manner (should be assigned to), which was confusing, thus we changed the text.“…A total of eight men participated in two, single-blind, repeated-measures trials. They were assigned to two hydration protocols, ingesting low mineralized table water and rich-bicarbonate water…” …” (Page 2, Lines 93-95). 

We thank the reviewer for all the comments, which we have addressed in the response hoping that the text has been significantly improved and can now be accepted for print in the IJERPH.

Reviewer 2 Report

Overall, this is a well-written manuscript. The authors should clarify the focus on judo rather than all combat sports. My biggest concern is that if all participants completed the protocol in the same order (table water then rich-bicarbonate water), it is difficult to distinguish between the effects of the rich-bicarbonate water and routine judo training.

Line 3, 13, 284, and 340: Replace “elite combat sport athletes” with “elite judo athletes”

Line 30: Remove “to 36 minutes in professional boxing” since the focus of this manuscript is on elite judo athletes.

Line 72: Write out the “high intensity anaerobic exercise” (HIAE) acronym the first time it is used.

Line 95: Change “6,5%” to “6.5%”

Line 103/Table 1: Since this was a repeated measures design, the term “experimental group” is confusing. It was the same 8 people, right?

Lines 109-110: Was the order of treatment the same (i.e., table water then rich-bicarbonate water) or randomized?

Lines 110-112: How do we know that the improved anaerobic performance is related to “rich-bicarbonate water ingestion” and not 3 weeks of training?

Line 122: To clarify, were all foods and beverages provided to participants? There was no free choice?

Line 215: Write out the “experimental group” (EG) acronym the first time it is used.

Line 218: Change “statue” to “state”

Line 295: Should “Anaerobic performance and lactate metabolism” be a subheading?

Author Response

Response to Reviewer 2

Comments and Suggestions for Authors

Overall, this is a well-written manuscript. The authors should clarify the focus on judo rather than all combat sports.

We appreciate this comment. In the revised manuscript, we have modified the original version based on the comments and suggestions.

Comment 1:

My biggest concern is that if all participants completed the protocol in the same order (table water then rich-bicarbonate water), it is difficult to distinguish between the effects of the rich-bicarbonate water and routine judo training.

Response 1:

At the beginning, the authors would like to mention that at the stage of constructing the experiment, several variants were analyzed and discussed. However, the single-blind, repeated measures trial model was chosen and considered to be justified. “…Eight male, elite judo athletes participated in two, single-blind, repeated-measures trials. They were assigned to two hydration protocols (Trials), ingesting low mineralized table water – pre-trial, and rich-bicarbonate water – trial…”                When attempting to justify the experimental model used, it should be emphasized that the main goal of the study is to compare selected anaerobic performance variables after exercise-induced dehydration. The participants of the experiment were elite athletes meeting the inclusion criteria, whose anaerobic performance is assessed in the state of hydration and dehydration (3% BM). Additionally, the rehydration time was observed. In this approach, the status of hydration, dehydration, and rehydration is a specific physiological state defined by weight loss and urine osmolality. Metabolic stress conditions, were achieved through: 1) hydration protocol 2) exercise leading to dehydration.

An important element of the research was to rule out the possibility that athletes are not optimally hydrated and that, as a consequence, the results of anaerobic performance assessment (double Wingate test protocol) in the state of hydration and subsequently in the state of exercise-induced dehydration will be false and unreliable.

Subsequently, the same participants - elite judo athletes - were subjected to two trials: 

  • PRE TRIAL- which is actually Preliminary Treatment. The purpose of this stage is the standardization of hydration conditions. Preliminary treatment included 21 days table water ingestion, re-evaluation of hydration status and qualification to the experimental protocol (exercise-induced dehydration + anaerobic performance tests + rehydration protocol).

   2)    TRIAL - the same scenario was used with rich-bicarbonate water ingestion. Two measurements: baseline and after 21 days of use rich-bicarbonate water were performed and analyzed.

Why did the authors use within-subjects designs model? First, because it is difficult to define table water as a therapeutic agent. Second, because the participants constituted an elite group of athletes with a limited number of participants. In the opinion of the authors, comparing the results of laboratory tests in the described procedure and stages of the experiment is fully justified and the methods of statistical analysis used should not be questioned.

The authors changed figure 1 to improve the readability of the study design …” (Page 4, Lines 114). We used the phrase “divided into” in an unfortunate manner (should be assigned to), which was confusing, thus we changed the text.“…A total of eight men participated in two, single-blind, repeated-measures trials. They were assigned to two hydration protocols, ingesting low mineralized table water and rich-bicarbonate water…” …” (Page 2, Lines 93-95).

 During the experiment all athletes were subjected to the same training loads.

“…The study was conducted during the preparatory period of the annual training cycle, when a high volume of work dominated the daily training loads…” (Page 3, Lines 111-113)

In addition, the dehydration group reported beneficial changes in the improvement of anaerobic performance after using high-bicarbonate water. Statistical analysis showed no statistically significant changes in participants anaerobic performance at optimal hydration. Thus, it is difficult to demonstrate the importance of training and its possible impact on the results of the Wingate Tests.

Comment 2:

Line 3, 13, 284, and 340: Replace “elite combat sport athletes” with “elite judo athletes”

Response 2:

We appreciate this comment. As suggested, we have changed the description of the participants.

Comment 3:

Line 30: Remove “to 36 minutes in professional boxing” since the focus of this manuscript is on elite judo athletes.

Response 3:

Thank you for your suggestion. The authors agree that the removed fragment will make it more specific and highlight the main topic which is judo. Changes have been made.

“…During competition, which may last from 4 minutes of regular time to even 7 minutes real time of the fight in judo, a decrease in repeated explosive power is primarily attributed to decompensated  acid-base disturbances, hypohydration and depletion of muscles glycogen [1]…” . (Page 1, Lines 28-31).

Comment 4:

Line 72: Write out the “high intensity anaerobic exercise” (HIAE) acronym the first time it is used.

Response 4:

Thank you for your comment. The abbreviation has been defined.

Comment 5:

Line 95: Change “6,5%” to “6.5%”

Response 5:

Thank you for your suggestion. Changes have been made.

Comment 6:

Line 103/Table 1: Since this was a repeated measures design, the term “experimental group” is confusing. It was the same 8 people, right?

Lines 109-110: Was the order of treatment the same (i.e., table water then rich-bicarbonate water) or randomized?

Lines 110-112: How do we know that the improved anaerobic performance is related to “rich-bicarbonate water ingestion” and not 3 weeks of training?

Response 6:

 The authors agree with the valuable comment of the reviewer – “…experimental group…” was replaced with “…experiment participants…”. The research was carried out in the within-subjects design model. All participants participated in all of the conditions of the experiment – pre-trial and Trial .

The revised manuscript, takes account of this information:

“…The research was carried out in the within-subjects design model. All participants participated in the first and second trial…”(Page 3, Lines 111-113).

Yes. The same hydration protocol was recommended. During both the pre-trial and trial (table water and high-bicarbonate water), the hydration of the participants was verified, excluding a significant factor that could affect the results of the Wingate test both under hydration and exercise-induced dehydration

 “…The volume of water intake was individualized based on the recommendation of the National Athletic Trainers Association and averaged 3.2 – 3.4 L per day…” (Page 5, Lines 136-137).

The authors understand the doubts of the reviewer, however they argue that the improvement in anaerobic performance is due to the use of water with a high-bicarbonate content, because:

1) there were no statistically significant changes in the improvement of anaerobic capacity under hydration conditions,

2) there was a statistically significant increase in HCO3 after trial  (rich-bicarbonate water ingestion) only,

3) in the group of elite athletes, a 3-week training period taking into account the implementation of the described training loads does not seem to be a strategic factor in improving Wingate Test performance, highlighting changes in dehydration conditions.

Comment 7:

Line 122: To clarify, were all foods and beverages provided to participants? There was no free choice?

Response 7:

Thank you for your comments. Meals for participants were prepared in the canteen in accordance with the energy demands and under the control of a dietitian. They all ate the same meals at the same time in accordance to the daily schedule. 

“…During the experiment the study participants lived in the dormitory and were fed at the academy cafeteria. The meals were prepared in the form of 24-h menus for seven days of the week. All meals were planed and supervised by a nutritionist. The quality and quantity of the food products was strictly controlled, maintaining proper proportions between the major macronutrients…”

Comment 8:

Line 215: Write out the “experimental group” (EG) acronym the first time it is used.

Response 8: As suggested by the reviewer, we have changed the abbreviations.

Comment 9:

Line 218: Change “statue” to “state”

Response 9:

Thank you for your suggestion. Changes have been made. The experimental group was replaced by participants.

Comment 10:

Line 295: Should “Anaerobic performance and lactate metabolism” be a subheading?

Response 10:

We appreciate this comment. Thanks for your suggestions. The changes have been made

We thank the reviewer for all the comments, which we have been addressed in the response hoping that the text has been significantly improved and can now be accepted for print in IJERPH.

Reviewer 3 Report

Dear Authors,

You have written an interesting study; however, some parts need to be addressed as there are several inconsistencies in the text.

First, there is your title. Your sample was comprised only of judo athletes so generalising your conclusions to all combat sports athletes is not ok as there are differences between striking and grappling combat sports. I recommend changing it to'' elite judo athletes''.

Add a keyword: combat sports

The introduction does not highlight how is the hydration measured and there is no rationale behind why did you choose your tests. Additionally, no connection on how is dehydration connected to performance to combat sports. Amend accordingly

Methods

Participants: First, how did you define that there were elite. Just from the fact that they are in the national team? Elaborate

Inclusion criteria: how did you come to the 12 years of experience inclusion criteria? Back this up with references

What were their judo level (dan grade) and training experience? report

You wrote that the sample size was 28. Explain. I assume from Polish under 81kg weight category? add the info

You wrote ''or were taking neuroactive or psychoactive drugs, stimulants, antioxidants or other illegal substances'' how did you check that? report

Screening tests

you wrote ''in order to meet the aforementioned selection criteria'' aforementioned WHERE? ADD the info and back up your decisions on inclusion criteria for vo2max and TW with references.

Table 2 - you wrote that '' three analysis of each type of water'' Where was this analysis carried out. Was the table water also controlled by you or was this from the participants home? Was this ''table water tested in their locations? Report

Experimental protocol

Report the ''medical weight scale'' model

What were the instructions and procedures before Bioimpedance measurements? report

Was the blood taken before or after the bioimpedance measurements? Report

You wrote, ''At 9:00 they were subjected to the dehydration protocol, which consisted of continuous running on a treadmill at an intensity of 50% VO2max'' and how was this Vo2 max determined? Were there any old results? Elaborate and report

You wrote'' while participants briefly stepped off the treadmill onto a floor scale '' Was the weight-adjusted for the running shoes and sports clothes or not? Report

''Under dehydration, cognitive tests were administered'' Really? You don't report any cognitive tests in the methods? Elaborate

How did you choose the rehydration criteria? Elaborate and back up your criteria with references

Where did you take the capillary blood? report

Anaerobic Performance:

just 3 minutes after a Wingate test? Elaborate? was this enough rest?

Warm-up - so 5 minutes for warm-up on both ergometers - How much time on each ergometer? It is not clear. Also, a 5 minute warm-up before a maximal all-out test? This is too short. Elaborate and back up your decision with references.

Decision on Wingate loads for upper and lower tests need additional references.

Why did you choose the incomplete rest intervals? What did you want to achieve with that? Elaborate

You report that you recorded the peak power; however, you don't report it anywhere in the tables. ADD

Figure 1. Flowchart - add days - elapsed time between stages for greater clarity

Results

Table 1. you report Vo2 max results, but you don't describe this in methods? Why? - add

Where are the results from cognitive tests and lower limbs Wingate in further analysis? Incomplete section

What is the practical application of your research? Add

In the end, the presentation of your paper is poor; however, the studies essence is interesting and this is the reason I am not rejecting this paper. I hope the authors will amend the paper accordingly.

Kind regards

Author Response

Response to reviewer 3

Comments and Suggestions for Authors

Dear Authors,

You have written an interesting study; however, some parts need to be addressed as there are several inconsistencies in the text.

We appreciate this comment. In the revised manuscript, we have modified the original version based on the comments and suggestions.

Comment 1:

First, there is your title. Your sample was comprised only of judo athletes so generalising your conclusions to all combat sports athletes is not ok as there are differences between striking and grappling combat sports. I recommend changing it to'' elite judo athletes''.

Add a keyword: combat sports

 Response 1:

We appreciate this comment. Thanks for your suggestions. The changes have been made (Page 1, Lines 3, 24).

Comment 2:

Introduction

The introduction does not highlight how is the hydration measured and there is no rationale behind why did you choose your tests. Additionally, no connection on how is dehydration connected to performance to combat sports. Amend accordingly

Response 2:

We thank for your suggestion. The indicated issue requires clarification. Information suggested by the reviewer has been added. The authors write as below: “…Various methods of assessing hydration status are used in hydration monitoring. These include total body water -bioimpedance, blood-osmolality, volume, isotope indexes, as well as sodium concentration, and urine osmolality, volume, urine specific gravity variables. In our study, we used Uosm and ultrasound monitoring. This choice is consistent with the suggestion of Popowski et al. [30], Who compared the validity of Uosm and USG with plasma osmolality and concluded that both are strongly correlated and are good measurements of the state of hydration…”(Page 2, Lines 48-54). 

Comment 3:

Methods

Participants: First, how did you define that there were elite. Just from the fact that they are in the national team? Elaborate

Inclusion criteria: how did you come to the 12 years of experience inclusion criteria? Back this up with references

What were their judo level (dan grade) and training experience? report

Response 3:

The authors thank you for your valuable comment. All of the above-mentioned issues require clarification. The choice of participants consequently influenced the choice of study design. The following adjustments were made:

„…All participants had at least twelve years of training experience, international sports level, and they were members of the Polish National Team. They were all medalists of the national championships and world cups. The participants constituted a homogenous group in regards to age, training experience, somatic characteristics, as well as aerobic and anaerobic fitness…” (Page 2, Lines 87-91).

Training experience was not a criterion for inclusion, but rather a factor exposing the elitism of the group and describing it. The authors, following the reviewer's suggestions, introduced some clarifications in this area.

“…The main criteria for inclusion were the medal of the National Championship and the starts in the World Cup in the last 24 months. Moreover, for the homogeneity of the research group, the authors used additional criteria, VO2max: higher than 58.0 mL/kg/min; TW–upper limbs; higher than 180 (J/kg), TW–lower limbs; higher than 220 (J/kg)…” (Page 4 , Lines 121-125).

Comment 4:

 You wrote that the sample size was 28. Explain. I assume from Polish under 81kg weight category? add the info

Response 4:

We thank for your suggestion. The indicated issue requires clarification. This text in the manuscript was inaccurate and incomplete. This excerpt showed the population size (28), not a representative sample of the population (n=8). The authors introduced changes and additions. “…To determine the sample size were used LEO Sound® at a 95% confidence level with a 6,5% margin of error and standard deviation, given that population size is 28. The sample size was estimated at n=8…” (Page 2, Lines 90-92).Additionally, in line with the reviewer’s suggestions, the authors emphasized the elitism of the participants and clarified the inclusion criteria. Estimates were made by the authors as below:

To determine the sample size were used LEOSound® at a 95% confidence level with a 6,5% margin of error and standard deviation, given that population size is 28, will be used formula:

where,

  • N = Population size,
  • Z = Critical value of the normal distribution at the required confidence level,
  • p = Sample proportion,
  • e = Margin of error

In this experiment

  • N = 28
  • Z = 1.95
  • p = 0.05 (uncertain sample proportion),
  • e = 6,5% or 0.065

n= (28 * (1.95 2)*0.05*(1-0.05)/(0.065 2)/(28 – 1+((1.95 2)* 0.05*(1-0.05)/(0.065 2))))=8

Sample Size = 8

Explanation of the Sample Size Formula

The calculation of the sample size formula has been done by using the following steps:

Step 1: Firstly, there were determine the population size which is the total number of distinct entities in population and it is denoted by N. [Note: N=28]

Step 2: Next, it was determined the critical value of the normal distribution at the required confidence level (critical value at 95% confidence level - 1.95)

Step 3: Next, it was determined the sample proportion which can be used from previous survey results or be collected by running a small pilot survey. [Note: p=0.05.]

Step 4: Next, it was determined the margin of error which is the range in which the true population is expected to lie. [Note: e=0.065]

Step 5: Finally, the sample size equation was derived by using population size (step 1), the critical value of the normal distribution at the required confidence level (step 2), sample proportion (step 3) and margin of error (step 4) as shown below.

The sample size was estimated at n=8.

“…To determine the sample size were used LEO Sound® at a 95% confidence level with a 6,5% margin of error and standard deviation, given that population size is 28. The sample size was estimated at n=8…” (Page 2, Lines 93).

The issue of the weight category also requires clarification. Added sentence:

“…The participants of the experiment were athletes competing in the weight category 81 kg (3 men) and 73 kg (5 men)…” (Page 2, Lines 89-91).

Comment 5:

You wrote ''or were taking neuroactive or psychoactive drugs, stimulants, antioxidants or other illegal substances'' how did you check that? report

Response 5:

Thank you for your valuable comment. Changes and additions were made as below.

“…The inclusion and exclusion criteria were assessed at the screening stage, based on an interview, health assessment and analysis of fitness test results…” (Page 4, Lines 126-128).

Comment 6:

Screening tests

You wrote ''in order to meet the aforementioned selection criteria'' aforementioned WHERE? ADD the info and back up your decisions on inclusion criteria for vo2max and TW with references.

Response 6:

We thank for your suggestion. The paragraph “Screening tests” has been reconstructed based on the reviewer's previous comments. Its current form includes clarifying the inclusion criteria and the characteristics of the research material.

“…All selected participants for the current study underwent preliminary tests and medical examinations before the intervention (pre-trial and trial). The main criteria for inclusion were the medal of the National Championship and the starts in the World Cup in the last 24 months. Moreover, for the homogeneity of the research group, the authors used additional criteria: VO2max: higher than 58.0 mL/kg/min; TW–upper limbs; higher than 180 (J/kg), TW–lower limbs; higher than 220 (J/kg). The participants were excluded if they reported a history of cognitive deficiencies, or were taking neuroactive or psychoactive drugs, stimulants, antioxidants or other illegal substances. The inclusion and exclusion criteria were assessed at the screening stage, based on an interview, health assessment and analysis of fitness test results. Six of the eight participants in the experiment had current performance tests (VO2max, Wingate Test) that were performed 3 weeks prior to the project. In the case of two participants, the diagnosis was carried out at the qualification stage, one week before the intervention. Afterwards, each study participant visited the laboratory to get acquainted with the research protocol…” (Page 4, Lines 11-133)

Comment 7:

Table 2 - you wrote that '' three analysis of each type of water'' Where was this analysis carried out. Was the table water also controlled by you or was this from the participants home? Was this ''table water tested in their locations? Report

Response 7:

               Thank you for the important question. Qualitative tests of water composition (High-bicarbonate water and table water) were carried out in a laboratory in Bulgaria. Table water was poured out in Georgia. Its composition was selected in the low mineralization state. Both table water and rich-bicarbonate water were provided to the athletes in identical size and shape of plastic bottles with no external label.

Comment 8:

Experimental protocol

Report the ''medical weight scale'' model. What were the instructions and procedures before Bioimpedance measurements? report

Response 8:

The medical scale on which the measurements were made was Seca, model 635, Germany. The information has been supplemented.

“…The measurements of body mass were performed on a medical scale with a precision of 0.1 kg (Seca, mo.635, Seca Co., Germany)…” (Page 5, Line 168)

Body composition was evaluated using the electrical impedance technique (Inbody 720, Biospace Co., Japan). The measurement procedures were performed according to the manufacturer's instructions. The day before, the participants had the last meal at 20.00. They reported to the laboratory after standardized breakfast, refraining from exercise for 48h. Body composition results were not analyzed. The measurement was treated as a second, control - confirming body weight

Comment 9:

Was the blood taken before or after the bioimpedance measurements? Report

Response 9:

Thank you for this question. Blood was drawn after body weight measurements (Bioimpedance). However, as mentioned above, the authors, aware of the planned protocol and rapid exercise-induced dehydration, did not make the body composition results (especially TBW%) meaningful. These data were not analyzed.

Comment 10:

You wrote, ''At 9:00 they were subjected to the dehydration protocol, which consisted of continuous running on a treadmill at an intensity of 50% VO2max'' and how was this Vo2 max determined? Were there any old results? Elaborate and report

Response 10:

We thank for your question. The indicated issue requires clarification.

As indicated in the reply to Comment 6, all participants had a VO2 max diagnosis that allowed to estimate exercise intensity for the dehydration protocol (50% VO2max).As suggested by the reviewer, the following information has been added to the section:

“…The ramp VO2max test was performed on a treadmill (H/P Cosmos, Pulsar, Germany), starting at a speed of 6 km/h, which was increased linearly (1km/h/1min) until volitional exhaustion. During the test, heart rate (HR), oxygen uptake (VO2), expired carbon dioxide (CO2), minute ventilation (VE), breath frequency (BF) and respiratory exchange ratio (RER) were measured continuously using the MetaLyzer 3B-2R spiroergometer (Cortex, Germany) in the breath-by-breath mode. The following two criteria were used to determine VO2max: (a) a plateau in VO2 despite an increase in running speed, (b) RER > 1.10…” (Page 7, Lines 222-228)

Comment 11:

You wrote'' while participants briefly stepped off the treadmill onto a floor scale '' Was the weight-adjusted for the running shoes and sports clothes or not? Report

Response 11:

The measurement was carried out in underwear.  The authors could not predict the weight change of clothing and shoes soaked in sweat. The control measurements did not last longer than 45 seconds. “…The measurement was carried out in underwear. The control measurements did not last longer than 45 seconds…”

Comment 12:

''Under dehydration, cognitive tests were administered'' Really? You don't report any cognitive tests in the methods? Elaborate

Response 12:

Thank you very much for noticing the error. Cognitive performance was indeed assessed in some participants, but these data are not analyzed in this manuscript. The fragment has been deleted.

Comment 13:

How did you choose the rehydration criteria? Elaborate and back up your criteria with references

Response 13:

Thank you for this question. The justification is a reply to comment 2:

“…Various methods of assessing hydration status are used in hydration monitoring. These include total body water -bioimpedance, blood-osmolality, volume, isotope indexes, as well as sodium concentration, and urine osmolality, volume, urine specific gravity variables. In our study, we used Uosm and ultrasound monitoring. This choice is consistent with the suggestion of Popowski et al. [30], Who compared the validity of Uosm and USG with plasma osmolality and concluded that both are strongly correlated and are good measurements of the state of hydration…”

Comment 14:

Where did you take the capillary blood? report

Response 14:

Thank you for your question. Blood was drawn from the fingertip. Description is in section 2.2.1. Lactate, acid-base balance, ion concentration:

…The measurements were performed from fingertip capillary blood samples at rest and after 3 minutes of post exercise recovery…” (Page 6, Lines 187-189)

Comment 15:

Anaerobic Performance

  1. just 3 minutes after a Wingate test? Elaborate? was this enough rest?
  2. Warm-up - so 5 minutes for warm-up on both ergometers - How much time on each ergometer? It is not clear. Also, a 5 minute warm-up before a maximal all-out test? This is too short. Elaborate and back up your decision with references.
  3. Why did you choose the incomplete rest intervals? What did you want to achieve with that? Elaborate
  4. Decision on Wingate loads for upper and lower tests need additional references.

Response 15:

Once again, thank you very much for your valuable insight and suggestions for clarifications.Let me systematize the answer to the above 4 points in 2 issues: exercise protocol and warm-up. Warm-upThe description of the warm-up is part of the protocol for special preparation. “…The test was preceded by 20 minutes general warm-up, as well as 5 min special warm-up. The preparations consisted of a 12-minute, low-intensity, continuous exercise on an elliptical cross trainer (Keiser Stride, m5i, USA) and 8 minutes of general calisthenic exercises. Then the participants performed special exercises on diagnostic ergometers - 5 minutes with a resistance of 100 W and a cadence range of 70-80 rpm for the lower limbs and 40 W and 50-60 rpm for the upper limbs…” (Page 7, Lines 211-215).  Exercise Protocol                The reviewed paper is an experiment in the research cycle on the effects of dehydration on anaerobic and cognitive performance, conducted by our team. The authors standardize the exercise protocol (single, double, quadruple, Wingate Test) to predict acute and extreme acid-base disturbances (using the potential of highly trained athletes). The repeatability of the protocols allows us to verify the hypotheses of the influence of the hydration protocols or the use of supplementation (e.g. with sodium bicarbonate) on anaerobic as well as  cognitive performance. The research project was carried out on a small, but representative study sample.              The authors justify the choice of the double Wingate test protocol in order to induce severe acid-base disturbances and experiences from previous research enable a broader perspective for comparing anaerobic performance variables. An incomplete rest between Wingate tests (upper +lower limbs) was to lead to extreme disturbances in the acid-base balance and extreme lactate efflux. The influence of alkaline water as well as sodium bicarbonates on acid-base balance on anaerobic performance using the modified Wingate test protocol was presented in the works: Alkaline water improves exercise induced metabolic acidosis and enhances anaerobic exercise performance in combat sport athletes. PLoS ONE 13(11): 

Anaerobic power and hydration status in combat sport athletes during body mass reduction Baltic Journal of Health and Physical Activity 2019;11(4):1-8.

  Chronic ingestion of sodium and potassium bicarbonate, with potassium, magnesium and calcium citrate improves anaerobic performance in elite soccer players. Nutrients Vol. 10, nr 11 (2018), s. 1-12.

Comment 16:

You report that you recorded the peak power; however, you don't report it anywhere in the tables. ADD

Where are the results from cognitive tests and lower limbs Wingate in further analysis? Incomplete section.

Response 16:

Due to the large number of assessed parameters, the authors in the original version of the manuscript limited themselves to presenting statistically significant changes. 

 In the revised version, changes to peak power have been added. (Page 8, Lines 268).

  As mentioned in the reply to Comment 12, the fragment relating to the cognitive performance assessment was a mistake by the authors and should not be included in the reviewed manuscriptWe apologize for inaccuracies

Comment 17:

Figure 1. Flowchart - add days - elapsed time between stages for greater clarity

 Response 17:

Thanks for your valuable suggestions. The authors changed figure 1 to improve the readability of the study design …” (Page 4, Lines 125).

Comment 18:

Results Table 1. you report Vo2 max results, but you don't describe this in methods? Why? - add

Response 18:

We thank for your question. The indicated issue requires clarification. As suggested by the reviewer, the following information has been added to the section “Anaerobic and Aerobic Performance”:

“…The ramp VO2max test was performed on a treadmill (H/P Cosmos, Pulsar, Germany), starting at a speed of 6 km/h, which was increased linearly (1km/h/1min) until volitional exhaustion. During the test, heart rate (HR), oxygen uptake (VO2), expired carbon dioxide (CO2), minute ventilation (VE), breath frequency (BF) and respiratory exchange ratio (RER) were measured continuously using the MetaLyzer 3B-2R spiroergometer (Cortex, Germany) in the breath-by-breath mode. The following two criteria were used to determine VO2max: (a) a plateau in VO2 despite an increase in running speed, (b) RER > 1.10…” (Page 7, Lines 222-228)

Comment 19:

What is the practical application of your research? Add

Response 19:

The practical application completed the conlusions sections.

“…Certainly, anaerobic performance is affected by the hydration status of the athletes. Both in the context of competitions and training routine, hydration is part of the strategy to reduce the possibility of uncompensated water loss.The presented results show the beneficial effect of high-bicarbonate water ingestion on buffering capacity, anaerobic fitness and the rate of rehydration. The athletes in the present study used rich-bicarbonate water to improve their buffering capacity, and we demonstrated the potential role of this mechanism and its phenomenon in masking the adverse effects of dehydration in the context of repeated HIAE…” (Page 11, Lines 392-400).

We thank the reviewer for all the comments, which we have been addressed in the response hoping that the text has been significantly improved and can now be accepted for print in IJERPH.

Round 2

Reviewer 1 Report

Thank you for the revised manuscript, however some of the initial concerns have not been adequately addressed.

The tables are still poorly formatted, and each table seems to have a different format. Please use centre margins and be consistent with the line thickness and try to have the data on one line. A reviewer should not have to request editing of tables in a second round review. Please address this properly.

The sample size calculation should be removed. The assumption of 6.5% margin is poorly justified. I did ask the authors to call this a case study or pilot study. The latter suggestion is perfectly acceptable given the elite nature of the participants. Trying to fudge a sample size calculation to fit the data is poor science.

The participant characteristics should include how much training experience the group has had, and current fitness status. The authors claim the participants are homogenous, but provide no data on fitness or training experience to justify this statement.

The article has merit and the second version is improved, but these outstanding issues need to be adequately addressed. 

Author Response

Response to reviewer 1 (ROUND 2)

Comments and Suggestions for Authors

  Thank you for the revised manuscript, however some of the initial concerns have not been adequately addressed.

The article has merit and the second version is improved, but these outstanding issues need to be adequately addressed. 

We appreciate this comment. In the revised manuscript, we have modified the original version based on the comments and suggestions.

Comment 1:

The tables are still poorly formatted, and each table seems to have a different format. Please use centre margins and be consistent with the line thickness and try to have the data on one line. A reviewer should not have to request editing of tables in a second round review. Please address this properly.

Response 1:

The authors apologize for this inconvenience, and we appreciate your understanding. Tables have been reformatted to meet the journal’s standards. All comments of the reviewer were taken into account. We hope they are acceptable in their current form.

Comment 2:

Materials and methods.

The sample size calculation should be removed. The assumption of 6.5% margin is poorly justified. I did ask the authors to call this a case study or pilot study. The latter suggestion is perfectly acceptable given the elite nature of the participants. Trying to fudge a sample size calculation to fit the data is poor science.

Response 2:

The authors agree to the reviewer's suggestions. The presented research, mainly due to the limited number of participants and demanding selection criteria, can be considered a pilot study. However, the methods of assessing anaerobic capacity, hydration and metabolic responses used by the authors are not a novelty, have not been modified, have not been validated and have been often used in previous studies.

The authors do not agree with the allegation that the calculation of the sample size was falsified. In the manuscript, we describe the rationale for the calculations - theoretical and related to the availability of elite athletes for testing.

However, to avoid potential doubts and allegations, the authors made changes as suggested by the reviewer. Thus the title has been changed as follows:

“…Chronic ingestion of rich-bicarbonate water improves anaerobic performance in hypohydrated elite judo athletes: A pilot study…” (Page 1, Lines 4).

This narrative is maintained in the text.

Comment 3:

The participant characteristics should include how much training experience the group has had, and current fitness status. The authors claim the participants are homogenous, but provide no data on fitness or training experience to justify this statement.

Response 3:

Thanks for this valuable comment. The participation of elite athletes in the project and their limited availability and number determined the methodology chosen in the project. The authors used a purposeful selection of participants, setting very strict and demanding criteria. They included chosen variables of aerobic and anaerobic performance, as well as training and competition experience. As suggested by the reviewers, these data were corrected and included in the methodology section of the manuscript. 

“…All selected participants for the current study underwent preliminary fitness tests and medical examinations before the intervention. The main criteria for selection included sports performance, based on success in the National Championships and the World Cup competition in the last 24 months. Moreover, for the homogeneity of the research group, the authors used additional criteria: VO2max: higher than 58.0 mL/kg/min; TW–upper limbs; higher than 180 (J/kg), TW–lower limbs; higher than 220 (J/kg)…”(Page 4, Lines 128-132).

“…All participants had at least twelve years of training experience, international sports level, and they were members of the Polish National Team. They were all medalists of the national championships and world cups. The participants of the experiment were athletes competing in the weight category 81 kg (3 men) and 73 kg (5 men)...” (Page 2, Lines 94-98).

The characteristics of the participants are also presented in Table 1 (Page 3, Lines 113).

Once again, we would like to thank the reviewer for all the valuable comments. We believe that the text has been significantly improved and hopefully the manuscript meet the standards of IJERPH, and can now be accepted for print.

Reviewer 2 Report

Thanks for your responsiveness to reviewer feedback. Overall, I believe that you have done a great job of responding to my comments.

General comment: Many words are used to describe the hydration protocols and this creates confusion. The “table water” protocol is also called “baseline,” “pre-trial,” and “first.” The “rich-bicarbonate water” protocol is also called “after,” “trial,” and “second.” For clarity, I believe it would be helpful to call these “table water” and “rich-bicarbonate water” throughout the text, tables, and figures.

Lines 49-51: With all the dashes and commas, it is hard to follow how these techniques fit together. Perhaps this could be simplified to say: “These include bioimpedance analysis, hematologic markers, isotope indices, and urinary markers.”

Line 94: Please clarify the meaning of “apparently.”

Line 100: “were” should be “we”

Line 101-102: Please clarify the population size of 28.

Table 1: For conciseness, the mean and standard deviation should be on the same line.

Line 120: “designes” should be “design” / “participate” should be “participated”

Line 139: “diagnosis was” should be “performance tests were”

Line 153: What were the macronutrient percentages/proportions?

Tables 3/4/5/6: The various superscripts (*#&) and bolding are confusing. My understanding is that all of these denote differences between “baseline” and “after,” which I believe would be better titled “table water” and “rich-bicarbonate water.”

Author Response

Response to Reviewer 2 (Round 2)

Comments and Suggestions for Authors

Thanks for your responsiveness to reviewer feedback. Overall, I believe that you have done a great job of responding to my comments.

We appreciate this comment. In the revised manuscript, we have modified the original version based on the comments and suggestions.

General comment:

Comment 1:

Many words are used to describe the hydration protocols and this creates confusion. The “table water” protocol is also called “baseline,” “pre-trial,” and “first.” The “rich-bicarbonate water” protocol is also called “after,” “trial,” and “second.” For clarity, I believe it would be helpful to call these “table water” and “rich-bicarbonate water” throughout the text, tables, and figures.

Response 1:

The authors agree with the reviewer's opinion. This multitude of descriptions used to describe the hydration protocols is indeed confusing. This was caused by the need to emphasize the study design, and in fact to distinguish between the state of hydration from evaluations  performed after ingestion of table water and rich-bicarbonate water. Nevertheless, the descriptions of the protocols have been changed throughout the text, tables and figures in accordance with the reviewers suggestions.

Comment 2:

With all the dashes and commas, it is hard to follow how these techniques fit together. Perhaps this could be simplified to say: “These include bioimpedance analysis, hematologic markers, isotope indices, and urinary markers.”

Response 2:

Thanks for your comment and suggestions. The text has been corrected as recommended. The   text should now be significantly improved and more understandable. 

“…These include electrical impedance analysis, hematologic markers, isotope indices, and urinary markers. In our study, we used urine osmolality (Uosm) and urine specific gravity (Ugrav)…”(Page 2, Lines 49-50).

Comment 3:

Line 94: Please clarify the meaning of “apparently.”

Line 100: “were” should be “we”

Line 120: “designes” should be “design” / “participate” should be “participated”

Line 139: “diagnosis was” should be “performance tests were”

Response 3:

We appreciate the precise study of the manuscript and appropriate comments. The above changes have been made. In addition, we would like to emphasize that the manuscript has once again been linguistically improved.

“…To determine the sample size we used the LEO Sound® at a 95% confidence level with a 6.5% margin of error and standard deviation, given that population size is 28. The sample size was estimated at n=8...” (Page 3, Lines 100).

“…The research was carried out in the within-subjects design model. All participants participated in first and second trials…”(Page 3, Lines 118).

“…Six of the eight participants in the experiment had performance tests (VO2max, Wingate Test) that were performed 3 weeks prior to the project. In case of two participants, the performance tests were carried out at the qualification stage, one week before the intervention. These discrepancies were caused by participation in competitions...” (Page 4, Lines 137).

Comment 4:

Line 101-102: Please clarify the population size of 28.

Response 4:

The size of the population includes members of the Polish National Judo Team only in two weight categories of 73 kg and 81 kg in the years 2019-2020. Additionally, the inclusion criteria such as VO2max, TW from the Wingate Test Protocol have been specified.

The following data has been included in the manuscript:

“…The size of the population includes members of the Polish National Judo Team in weight categories of 73 kg and 81 kg in 2019-2020. Only these weight categories were included, to create a homogenous group of participants. Including more athletes from extreme weight categories (under 60 or above 100kg) would make the hydration protocols very difficult…”. (Page 4, Lines 137)

“…The main criteria for selection included sports performance at the National Championships and   the World Cup in the last 24 months. Moreover, for the homogeneity of the research group, the authors used additional criteria, VO2max: higher than 58.0 mL/kg/min; TW–upper limbs; higher than 180 (J/kg), TW–lower limbs; higher than 220 (J/kg)…” (Page 4 , Lines 121-125).

Comment 5:

Table 1: For conciseness, the mean and standard deviation should be on the same line.

Tables 3/4/5/6: The various superscripts (*#&) and bolding are confusing. My understanding is that all of these denote differences between “baseline” and “after,” which I believe would be better titled “table water” and “rich-bicarbonate water.”

Response 5:

Tables have been reformatted. We hope their readability has improved.

Comment 6:

Line 153: What were the macronutrient percentages/proportions?

Response 6:

The text has been corrected according to the suggestions.

“…The participants were placed on an isocaloric (3455±436 kcal/day) mixed diet (55% carbohydrates, 20% protein, 25% fat) prior to and during the investigation…”(Page 5 , Lines 149-151).

Once again, we would like to thank the reviewer for all the valuable comments. We believe that the text has been significantly improved and hopefully the manuscript meet the standards of IJERPH, and can now be accepted for print.

Reviewer 3 Report

Dear Authors,

Thank you for addressing the raised questions as this makes your study more reproducible. Overall in my opinion the paper is in acceptable form for publication which still needs some minor spelling check.

Kind regards

Author Response

Response to reviewer 3

Comments and Suggestions for Authors

Dear Authors,

Thank you for addressing the raised questions as this makes your study more reproducible. Overall in my opinion the paper is in acceptable form for publication which still needs some minor spelling check.

Once again we would like to thank the reviewer for his time and dedication, while reviewing our manuscript and making valuable suggestions. We have revised the paper to the best of our knowledge and made significant corrections in grammar and English style.
